# Protein Quality in Infant Formulas Marketed in Brazil: Assessments on Biodigestibility, Essential Amino Acid Content and Proteins of Biological Importance

**DOI:** 10.3390/nu13113933

**Published:** 2021-11-03

**Authors:** Cristine Couto de Almeida, Diego dos Santos Baião, Katia Christina Leandro, Vania Margaret Flosi Paschoalin, Marion Pereira da Costa, Carlos Adam Conte-Junior

**Affiliations:** 1Graduate Program in Sanitary Surveillance (PPGVS), National Institute of Health Quality Control (INCQS), Oswaldo Cruz Foundation (FIOCRUZ), Rio de Janeiro 21040-900, Brazil; almeidacristine@hotmail.com (C.C.d.A.); kcleandro@gmail.com (K.C.L.); 2Center for Food Analysis (NAL), Technological Development Support Laboratory (LADETEC), Federal University of Rio de Janeiro (UFRJ), Cidade Universitária, Rio de Janeiro 21941-598, Brazil; 3Graduate Program in Veterinary Hygiene (PPGHV), Faculty of Veterinary Medicine, Fluminense Federal University (UFF), Vital Brazil Filho, Niterói 24230-340, Brazil; marioncosta@id.uff.br; 4Laboratory of Advanced Analysis in Biochemistry and Molecular Biology (LAABBM), Department of Biochemistry, Federal University of Rio de Janeiro (UFRJ), Cidade Universitária, Rio de Janeiro 21941-909, Brazil; diegobaiao20@hotmail.com (D.d.S.B.); paschv@iq.ufrj.br (V.M.F.P.); 5Graduate Studies in Food Science (PPGCAL), Institute of Chemistry (IQ), Federal University of Rio de Janeiro (UFRJ), Cidade Universitária, Rio de Janeiro 21941-909, Brazil; 6Graduate Studies in Chemistry (PGQu), Institute of Chemistry (IQ), Federal University of Rio de Janeiro (UFRJ), Cidade Universitária, Rio de Janeiro 21941-909, Brazil; 7Laboratory of Inspection and Technology of Milk and Derivatives (LaITLácteos), School of Veterinary Medicine and Animal Science, Federal University of Bahia (UFBA), Salvador 40170-110, Brazil

**Keywords:** breastfeeding, infant formulas, protein quality, whey proteins, caseins, HPLC, amino acid score, amino acid score corrected protein digestibility

## Abstract

Infant formulas, designed to provide similar nutritional composition and performance to human milk, are recommended when breastfeeding is not enough to provide for the nutritional needs of children under 12 months of age. In this context, the present study aimed to assess the protein quality and essential amino acid content of both starting (phase 1) and follow-up (phase 2) formulas from different manufacturers. The chemical amino acid score and protein digestibility corrected by the amino acid score were calculated. The determined protein contents in most formulas were above the maximum limit recommended by FAO and WHO guidelines and at odds with the protein contents declared in the label. All infant formulas contained lactoferrin (0.06 to 0.44 g·100 g^−1^) and α-lactalbumin (0.02 to 1.34 g·100 g^−1^) below recommended concentrations, whereas ĸ-casein (8.28 to 12.91 g·100 g^−1^), α-casein (0.70 to 2.28 g·100 g^−1^) and β-lactoglobulin (1.32 to 4.19 g·100 g^−1^) were detected above recommended concentrations. Essential amino acid quantification indicated that threonine, leucine and phenylalanine were the most abundant amino acids found in the investigated infant formulas. In conclusion, infant formulas are still unconforming to nutritional breast milk quality and must be improved in order to follow current global health authority guidelines.

## 1. Introduction

Infant nutrition during the first two years of life is essential for adequate development and crucial for survival and long-term health and well-being. Nourishment deficiencies caused by inadequate nutrition during this development stage may cause immediate damage, increasing infant morbidity and mortality, while also potentiating growth delays and low school achievements and increasing the risks for chronic and degenerative adulthood diseases [1,2]. In this regard, breast milk is the complete and ideal infant nutritional source, endorsed by health authorities and recommended as the exclusive food source during the first six months of life, aiming at optimal child growth, development, and health. After sixth months, adequate and safe complementary feeding should be given to infants in order to meet their evolving requirements, while breastfeeding should be maintained at least until two years of age [3,4,5,6]. Breast-milk substitutes are, however, required when breastfeeding is not possible or recommended or if it does not meet nutritional needs. In this regard, infant formulas (IFs) are the only acceptable milk derivative for infants under one year old, as they can be manipulated to fulfill physicochemical and nutritional characteristics while still maintaining a composition as similar as possible to breast milk [5,6].

IFs are available in both liquid or powdered form and contain proteins isolated from cow milk or other the milk of animal species, as well as from vegetables, in both intact or hydrolyzed forms, although other nutrients may be added in the amounts and ratios recommended for each child age to ensure adequate growth, optimal development and immune and metabolic system maturation [7,8,9]. Researchers in the infant nutrition field and IFs manufacturers have developed a variety of IFs throughout the last decades, currently available worldwide and developed based on scientific infant nutrition data. IFs are formulated for the first months of life until the introduction of complementary feeding, and from the sixth month onwards, until age one. In addition to traditional formulas, specials IFs can also be formulated to provide specific dietary needs required in special health conditions [3,6].

IFs formulated from cow milk are produced at low-costs and widely marketed, and processed to adjust macro and micronutrient contents to obtain nutritional characteristics as similar as possible to human breast milk, which is safer for neonates or infants under one year old [10]. Cow milk processing aims to reduce protein content to avoid overloading the newborn immature renal tubular systems, mainly through the reduction of casein content and consequent improvement of whey protein: casein ratios in order to enrich IFs with high biological-activity proteins aiming to supply essential amino acids and facilitate milk digestion [1]. Furthermore, the quality and three-dimensional conformation of these proteins are essential to sustain infant safe growth and development, aiming at long-term health, as adequate protein intake during the first two years of life leads to important muscle protein syntheses and linear growth effects, as well as healthy immune and digestive system development and optimal brain development support, including better cognitive evolution [11,12].

Despite the attempts of manufacturers to mimic the composition and/or performance of human milk, it is not yet possible to state that IFs nutrients display the same bioavailability as human milk components since differences between the development and long-term health of infants fed IFs compared to exclusively breastfed individuals are still significant [13]. Furthermore, considering that IFs are the only food source for several neonates and infants, and that protein requirements during initial child developmental stages are the highest, the quality of proteins in marketed IFs should be monitored. In this context, the aim of the present study was to evaluate several IFs brands marketed in Brazil recommended for children from 0 to 6 months of age (starting formulas, phase 1) and for children aged 6 to 12 months (follow-up, phase 2) formulas with regard to protein quality through a combination of different analytical protein determination methods. To this end, total protein content, distribution and profiles and essential amino acid contents were determined and the results were then used to calculate the chemical amino acid score (AAS) and protein digestibility corrected by the amino acid score (PDCAAS).

## 2. Material and Methods

### 2.1. Sample Selection

IFs were purchased in commercial establishments located in the metropolitan region of the municipality of Rio de Janeiro, Brazil. All selected IFs are marketed in several commercial establishments, conveniently available to be purchased by consumers in the most populated part of the city. Although no formal inquiry was performed, as the selected formulas are commonly supplied in the largest supermarkets in town, they are considered a high demand by consumers and were, thus, evaluated in the present study.

All products are registered by the Brazilian regulatory agency ANVISA, marketed as powdered formulas packed in aluminum cans and labeled by each manufacturer. Inclusion criteria consisted of the following: formulas should be prepared exclusively from non-hydrolyzed cow milk proteins, for children aged 0 to 6 months (phase 1 starting IFs) and for children aged 6 to 12 months (phase 2 follow-up IFs). IFs produced using protein sources other than cow milk, such as soy or wheat protein, as well those designed for specific needs, such as lactose-free or hydrolyzed protein conditions, were not selected. A total of ten formulas marketed in Brazil produced by three different manufacturers filled the inclusion criteria. Thus, five phase 1 and five phase 2 formulas, and three distinct batches of each brand were thus selected, totaling thirty samples (N = 30). The three different batches of each brand were identified using the same two capital letters and a number, where the number indicates different batches, from 1 to 3. Manufacturers and brands were not disclosed for ethical reasons, and samples were identified by codes (Appendix A)

### 2.2. Total Protein Contents

Total protein contents were assessed according to the Kjeldahl method proposed by the Association of Official Analytical Chemists [14]. Nitrogenous materials were converted into protein content by employing a 6.25-factor multiplication.

### 2.3. Protein Fraction Analysis

#### Sodium dodecyl sulfate-polyacrylamide gel electrophoresis (SDS-PAGE)

A protein electrophoretic analysis was carried out in six stages, namely extraction, quantification, preparation and SDS-PAGE resolving followed by gel staining and destaining, and, finally, photo documentation [15]. Protein extraction was performed according to Almeida et al. [16] with modifications. First, proteins were quantified by serial dilution (1:1, 1:10 and 1:100) using Coomassie Blue G-250^®^ at 595 nm and 450 nm using a UV-1800 spectrophotometer (Shimadzu^®^, KYO, JPN). After establishing the appropriate dilution factor, the samples were diluted in the staining solution (4% SDS, 0.5 M Tris-HCl pH 6.8, 50 mM DTT, 20% glycerol and a pinch of bromophenol blue) and a total of 20 μL of each sample (5 mg·mL^−1^ of protein) were loaded in each well of 4% and 12% stacking and resolution acrylamide gels, respectively, and the runs were performed at 200 V and 25 mA for about 2 h. Apparent molecular weights were estimated using electrophoresis molecular marker weights (Precision Plus ProteinTM Standards, Bio-Rad) applied to the central well. After the runs, the gels were stained with Coomassie Blue G-250 for 24 h, destained with 20% acetic acid, 20% methanol, and 60% distilled water to visualize the protein bands and photo documented using a Gel Doc XR + Gel Documentation System. The relative abundance of the gel bands was estimated by TotalLab Quant^®^ software (TotalLab Ltd., Newcastle-Upon-Tyne, UK), through staining intensity and thickness.

### 2.4. High-Performance Liquid Chromatography (HPLC)

The identification and quantification of the main target proteins in the investigated IFs, α-lactalbumin (α-LA), β-lactoglobulin (β-LG), Κ-casein (Κ-CN), α-casein (α-CN), β-casein (β-CN) and lactoferrin (Lf), were performed by HPLC, following protein extraction according to Bobe et al. [17]. Briefly, about 10 g of each IFs were diluted in 100 mL of Milli-Q^®^ water (Merck Millipore, MA, USA) and 500 μL were mixed to 500 μL of a solution containing 0.1 M bi-tris-HCL/buffer (pH 6.8), 6 M guanidine hydrochloride, 5.37 mM sodium citrate and 19.5 mM d-dithiothreitol (pH 7.0). The resulting suspensions were incubated for 1 h at room temperature and then centrifuged for 20 min at 20,000× *g* at 4 °C. The small lipid layer formed during this step was removed, and the remaining solubilized solution was diluted 1: 3 (v:v) with 4.5 M guanidine hydrochloride and solvent A, and maintained at −20 °C until analysis.

The HPLC system comprised a quaternary pump LC-20AD (Shimadzu^®^, KYO, JPN), coupled to an analytical column C18 (250 mm × 4.6 mm, I.D., Kromasil^®^), and a photodiode array detector model SPD-20A (Shimadzu^®^) managed by the LabSolutions System software (Shimadzu^®^), employed in two distinct chromatographic conditions: (*i*) Κ-CN, α-CN, α-La, β-CN and β-Lg separation and identification according to Bonfatti et al. [18] at 214 nm employing an elution at a flow rate set at 0.5 mL·min^−1^ using a 0.1% trifluoroacetic acid (TFA) in water, and 0.1% TFA in acetonitrile mixture. An injection volume of 20 µL was applied, the total analysis time for each sample was 45 min and the column temperature was maintained at 45 °C; and (*ii*) lactoferrin separation and identification as described by Duchén et al. [19] at 205 nm employing a linear gradient and flow rate of 1 mL·min^−1^ and the previously described mobile phase. An injection volume of 10 µL was applied, the total analysis time for each sample was 25 min and the column temperature was set at 45 °C.

Major bovine milk protein standards were prepared in acetonitrile, water, and TFA at a 100:900:1 (v:v:v) ratio. Calibration curves were constructed by plotting increasing protein standard concentrations, and all samples and standards were run in triplicate.

### 2.5. Essential Amino Acid Identification

Nine essential amino acids threonine, lysine, histidine, valine, methionine, isoleucine, leucine, phenylalanine and tryptophan were identified and quantified in the IFs samples according to Furota et al. [20], with modifications. Phase 1 and phase 2 IFs samples were suspended in Milli-Q^®^ water (Merck Millipore) and 10 mL of each sample were mixed with 10 mL of 6N HCl and heated in an oven at 110 °C for 24 h. Hydrolyzed samples were then transferred to 50 mL volumetric flasks, made up with Milli-Q^®^ water and triplicate aliquots were filtered through non-sterile 0.22 μm MF-Millipore^®^ hydrophilic membranes (Millipore, MA, USA). The filtered solutions were maintained at −20 °C until analysis.

A Thermo Scientific^TM^ UltiMate^TM^ 3000 RSLC nano System (Thermo Fisher Scientific^®^, CA, USA) liquid chromatograph equipped with a Corona Ultra Charged Aerosol Detector (CAD) (Corona Veo, Thermo Scientific^®^) was used. Orthogonal chromatography was performed using cyan columns (150 mm × 4.6 mm, I.D., Phenomenex) connected to a C8 column (150 mm × 4.6 mm, I.D., Phenomenex). Data were acquired with the Chromeleon 7.2 software (Thermo Fisher Scientific^®^).

Essential amino acids were eluted at a 0.5 mL·min^−1^ flow rate using A 0.1% formic acid (FOA) in water and solvent B 100% acetonitrile (ACN) mixture. The gradient elution (solvent B) was increased from 0% to 25% from 0 to 30 min; 25% to 70% from 31 to 46 min and 70% to 0% from 47 to 60 min. A 20 µL injection volume was applied, the column temperature was maintained at 30 °C and the total analysis time for each sample was 60 min. Amino acid standards were purchased from Sigma (Sigma-Aldrich Co, MO, USA) ranging from 50 to 63 mg·100 mL^−1^.

### 2.6. In Vitro Gastrointestinal Digestion Simulation

An in vitro gastrointestinal digestion mimicking physiological conditions during the gastric and intestinal phases was performed according to Oomen et al. [21] and Sagratini et al. [22], with modifications.

### 2.7. Gastric Phase Digestion (GPD)

In a glass vial, 5 g of each sample were mixed with 2 mL of Milli-Q^®^ water and 2.5 mL of artificial gastric fluid, containing 2.75 g NaCl; 0.27 g NaH_2_PO_4_; 0.82 g KCl; 0.42 g CaCl_2_; 0.31 g NH_4_Cl; 0.65 g glucose; 0.085 g urea; 3 g mucine; 2.64 g swine gastric pepsin; 1 g bovine albumin and 8.3 mL HCl. After adjusting the volume to 500 mL, the pH was adjusted to 2.0 with 5 M HCl, the vials were sealed with a rubber septum and the atmosphere was charged with N_2._ The vials were then transferred to an orbital shaker at 260 rpm and 37 °C for 2 h to complete the gastric phase digestion (GPD), followed by the removal of 5 mL aliquots for further analysis.

### 2.8. Intestinal Phase Digestion (IPD)

In this stage, 0.9 M NaHCO_3_ were added to the GPD mixture to adjust the pH to 6.0, followed by 2 mL of an artificial intestinal fluid containing 6.75 g NaCl; 0.517 g KCl; 0.205 g CaCl_2_; 3.99 g NaHCO_3_; 0.06 g KH_2_PO_4_; 0.0375 g MgCl_2_; 0.1375 g urea; 25 g swine bile; 4 g swine pancreatin; 1.2 g albumin bovine, 0.185 mL HCl and volume adjustment to 500 mL. The glass vials were then resealed under an N_2_ atmosphere and maintained in the same conditions as the GDP followed by the collection of 5 mL aliquots for further analysis.

### 2.9. In Vitro Protein Digestibility (IVPD) Assay

The supernatants collected at the end of representative in vitro digestions (gastric and intestinal) were analyzed regarding nitrogen content employing the Kjeldahl AOAC 930.29 method [14]. The IVPD values were calculated according to the following equation:$$\%\;\mathrm{Digestibility}=\frac{\mathrm{Ns}\;–\;\mathrm{Nb}}{\mathrm{Ns}\;\times \;100}$$
where Ns and Nb are the nitrogen content of the samples and blanks, respectively.

### 2.10. Amino Acid Score (AAS) and Protein Digestibility Corrected Amino Acid Score (PDCAAS)

To calculate the AAS, the essential amino acid concentrations in the IFs were estimated and compared with daily amino acid infant requirements using the amino acid content in breast milk proteins as reference to define amino acid scores for infant foods [23]. The PDCAAS was calculated by multiplying the AAS value of each essential amino acid by the protein digestibility score.

### 2.11. Statistical Analyses

Significant differences in IFs protein contents were assessed through a One-way analysis of variance (ANOVA) with repeated measures. Differences in protein fractions and amino acids between IFs and their respective batches were evaluated by a two-way analysis of variance (ANOVA) with repeated measures. Differences in phase 1 and phase 2 IFs IVPD, AAS and PDCAAS were estimated by a one-way analysis of variance (ANOVA) with repeated measures. An additional post hoc analysis (Bonferroni correction) was performed when a significant *F* was found. Results were considered significant when *p* < 0.05. Data were expressed as the means ± standard deviations (SD). All statistical analyses were carried out using the Graphpad Prism software version 5 for Windows^®^ (GraphPad Software, CA, USA). All measures were acquired in triplicate.

## 3. Results

### 3.1. Crude Protein Contents

Figure 1 displays the comparison between the total protein content of whole cow milk (CM) and phase 1 and phase 2 IFs. Whole CM total protein content was 32.47 ± 0.33 g·100 g^−1^, higher than all phase 1 and phase 2 IFs. Phase 2 formulas, termed ME2, DM2 and DA2, contained the highest total protein levels, 22.64 ± 0.56 g·100 g^−1^, 19.95 ± 0.71 g·100 g^−1^, 19.33 ± 1.20 g·100 g^−1^, respectively, when compared to the remaining phase 2 and all phase 1 formulas. NC1, NN1 and ME1 contained the lowest total protein contents, of 9.67 ± 0.03 g·100 g^−1^, 11.94 ± 0.61 g·100 g^−1^ and 11.81 ± 0.35 g·100 g^−1^, respectively, compared to the other IFs (*p* < 0.05).

### 3.2. Protein Fractions Analysis

#### Protein Profiles and Relative Protein Abundance

The SDS-PAGE profiles presented in Figure 2 displayed intense bands with apparent molecular masses similar to lactoferrin (80 kDa), α-casein (23 kDa), β-casein (27 kDa), κ-casein (19 kDa), β-lactoglobulin (18 kDa) and α-lactalbumin (14.2 kDa) in all analyzed IFs. Other bands that could correspond to whey proteins, such as bovine serum albumin (68 kDa), were also identified in the NN, NC and ME formulas.

According to the apparent volume determined by the TotalLab Quant^®^ software, the relative percentage of the protein fractions identified in the analyzed gels, they exhibited different patterns among IFs brands and between batches (Figure 3). Bands identified as caseins ranged from 22.8% to 47%, with 40.6% quantified in DA1; 47.0% in DA2; 29.8% in ME1; 40.7% in ME2; 22.8% in NC1; 27.2% in NC2; 34.4% in NN1 and 35.1% in NN2, while bands identified as whey proteins displayed high variations, ranging from 53.3% to 77.2%, totaling 59.4% in DA1; 53.0% in DA2; 70.2% in ME1; 59.3% in ME2; 77.2% in NC1; 72.8% in NC2; 65.6% in NN1 and 64.9% in NN2. Regarding whey proteins, β-Lg was the most abundant fraction, although numerous other minor proteins were also observed. In addition to proteins in different IFs that were markedly identified in the gels, the TotalLab Quant^®^ software also indirectly revealed other minor proteins.

Major protein fractions were identified and quantified in phase 1 and phase 2 formulas.

The identified CM proteins eluted in the following order: κ-CN, α-La, α-CN, β-CN and β-Lg. The expected separation of major caseins and whey proteins was achieved and their similarity to previously established retention times for major peaks allowed for the identification of protein fractions in every sample, as displayed in a representative chromatogram exhibited in Appendix A. Calibration curves derived from the calculated regression parameters for increasing concentrations of standard individual CM proteins were used to quantify caseins and β-Lactoglobulin (Appendix A). The six fractions were well separated, displaying good peak resolution, i.e., sharpness, and symmetry. Repeatability was observed for multiple measurements for each sample, with a relative standard deviation (RSD) ranging from 0.16 to 0.92% for retention times and from 1.01 to 5.02% for peak areas. Regarding the limit of detection (LOD) and limit of quantification (LOQ), proteins ranged from 0.01 to 0.15 mg·L^−1^ and 0.28 to 1.02 mg·L^−1^, respectively, and protein recovery ranged from 88% to 103.4%.

Average values for major phase 1 and phase 2 IFs protein fractions are presented in Table 1. The detected protein fractions from phase 1 were different from those detected in phase 2 IFs, and protein fractions from distinct formulas from the same phase were also significantly different. IFs with the highest Lf contents comprised DA2, with 3.80 ± 3.01 mg·g^−1^; ME1, 1.37 ± 0.15 mg·g^−1^; DM1, 1.37 ± 0.10 mg·g^−1^ and DA1, with 1.13 ± 0.15 mg·g^−1^. The highest α-CN contents were found in NC1, of 21.17 ± 2.74 mg·g^−1^, while ME2 contained 18.50 ± 0.81 mg·g^−1^; ME1, 14.77 ± 2.02 mg·g^−1^ and DA2, 14.71 ± 2.55 mg·g^−1^. The highest β-CN contents were detected in DM1, at 25.16 ± 3.37 mg·g^−1^; DM2, at 28.7 ± 3.96 mg·g^−1^ and in NN2, at 22.23 ± 3.62 mg·g^−1^. The highest κ-CN contents were NN1, 124.50 ± 3.37 mg·g^−1^; DA1, 106.53 ± 2.72 mg·g^−1^; ME2, 127.1 ± 3.90 mg·g^−1^; NN2, 115.53 ± 2.21 mg·g^−1^ and DA2, 107.17 ± 1.14 mg·g^−1^, and the highest β-Lg contents were observed in NN1, 39.83 ± 1.79 mg·g^−1^; in NN2, 32.61 ± 3.91 mg·g^−1^ and in ME2, 26.27 ± 3.17 mg·g^−1^. The highest α-La contents were observed in NC1, 12.10 ± 1.21 mg·g^−1^; in NN1, 13.23 ± 0.41 mg·g^−1^ and in NN2, 11.97 ± 0.87 mg·g^−1^.

Whole powdered bovine milk samples were analyzed to compare phase 1 and phase 2 IFs CM protein fractions. CM contained the highest α-CN (43.26 ± 0.30 mg·g^−1^), β-CN (57.95 ± 0.50 mg·g^−1^), κ-CN (125.14 ± 0.39 mg·g^−1^) and β-Lg (56.62 ± 0.41 mg·g^−1^) contents compared to all IFs, but the CM the lowest Lf (0.31 ± 0.02 mg·g^−1^) compared to phase 1 and phase 2 IFs.

Variation of major protein fractions between the three batches of the same phase 1 and phase 2 IFs brands. Table 2 displays the average major protein fractions values from three phase 1 and phase 2 IFs batches. In general, a significant difference in protein fractions was observed among different batches from the same manufacturer, except for Lf in NC1, NN1, DM1, DA1, ME2, NC2 and NN2 and β-CN in NN1, DA1 and NC2. The same was noted for α-La in ME1, ME2 and DA2, revealing protein fraction homogeneity among the different batches of these IFs. A significant difference (*p* < 0.001) between phase 1 and phase 2 formula brands was noted concerning the average value of each protein fraction. Phase 1 mean values ranged from 86.07 to 129.85 mg·g^−1^ for κ-CN, from 7.51 to 22.83 mg·g^−1^ for α-CN, from 0.31 to 13.76 mg·g^−1^ for α-La, from 7.54 to 29.19 mg·g^−1^ for β-CN, from 13.25 to 41.94 mg·g^−1^ for β-Lg and from 0.22 to 1.51 mg·g^−1^ for Lf, while, the means of phase 2 IFs ranged from 82.97 to 129.59 mg·g^−1^ for κ-CN, from 7.09 to 19.22 mg·g^−1^ for α-CN, from 0.51 to 12.57 mg·g^−1^ for α-La, from 4.72 to 33.10 mg·g^−1^ for β-CN, from 15.59 to 36.32 mg·g^−1^ for β-Lg and from 0.28 to 8.55 mg·g^−1^ for Lf. Thus, κ-CN was the most abundant protein, and α-La and Lf, the less abundant ones.

### 3.3. Amino Acid Quantification in Phase 1 and Phase 2 IFs

The average amino acid content in phase 1 and phase 2 IFs are presented in Table 3. Histidine ranged from 0.11 to 0.16 mg·g^−1^ and tryptophan, from 0.10 to 0.14 mg·g^−1^, and similar contents for both amino acids were observed in both phase 1 and phase 2 samples. DM1, however, presented the highest valine contents, of 1.34 ± 0.02 mg·g^−1^, as well as me threonine, 1.99 ± 0.01 mg·g^−1^; isoleucine, 1.69 ± 0.01 mg·g^−1^ and phenylalanine, 3.24 ± 0.01 mg·g^−1^, when compared to other phase 1 IFs. DA1 and DM1 presented the highest threonine and leucine contents, of 4.34 ± 0.01 and 4.19 ± 0.02 mg·g^−1^ and 4.14 ± 0.02 and 3.96 ± 0.02 mg·g^−1^, respectively, when compared to other phase 1 IFs. Furthermore, the highest lysine contents were found in NN1, NC1 and ME1, of 0.26 ± 0.03, 0.20 ± 0.03 and 0.20 ± 0.04 mg·g^−1^, respectively. Concerning phase 2 formulas, DM2 contained the highest threonine content, 4.70 ± 0.02 mg·g^−1^, as well as methionine, 2.13 ± 0.02 mg·g^−1^; isoleucine, 1.99 ± 0.01 mg·g^−1^; leucine, 4.56 ± 0.01 mg·g^−1^ and phenylalanine 3.28 ± 0.02 mg·g^−1^ compared to other phase 2 IFs. The highest lysine contents were observed in DM2 and NC2, 0.19 ± 0.01 and 0.19 ± 0.01 mg·g^−1^, respectively, and the highest valine values, of 1.29 ± 0.03, 1.27 ± 0.01 and 1.25 ± 0.01 mg·g^−1^ were detected in NC2, NN2 and DM2, respectively.

When comparing phase 1 vs. phase 2 IFs amino acid contents, a significant difference in phase 1 lysine, valine and isoleucine contents was observed, higher in phase 1 compared to phase 2. On the other hand, higher threonine, methionine, leucine and phenylalanine were detected in phase 2 IFs. Finally, histidine and tryptophan contents were similar in both types of formula (Table 3).

### 3.4. Phase 1 and Phase 2 IFs %IVPD

Phase 1 and phase 2 IFs %IVPD are displayed in Figure 4. After gastric digestion, ME1, NC1, NN1 and DM1 presented higher %IVPD values than DA1. Concerning intestinal digestion, ME1, NN1 and DM1 exhibited higher %IVPD than NC1 and DA1. Among phase 2 IFs, ME2 and NN2 presented the highest %IVPD when compared to NC2, DM2 and DA2 after gastric digestion. However, the %IVPDs after intestinal digestion were not significantly different when comparing all phase 2 IFs.

### 3.5. Phase 1 and Phase 2 IFs AAS and PDCAAS

Chemical amino acid scores (AAS) were calculated as recommended by FAO/WHO (2007), considering the ratio between essential amino acid contents and those recommended for children aged 0 to 12 months old, used to identify infant formulas that could limit essential infant amino acid supplies (Table 4). The AAS calculation indicated that all essential amino acids in phase 2 IFs presented a chemical score below 1.0. Similar results were also observed for phase 1 IFs, except for threonine in ME1, NC1, DM1 and DA1, where the AAS was slightly higher than 1.0, as follows: 1.011 ± 0.005 mg·g^−1^, 1.019 ± 0.010 mg·g^−1^, 1.008 ± 0.007 mg·g^−1^ and 1.169 ± 0.002 mg·g^−1^, respectively. Almost all amino acids presented higher AAS values in phase 1 compared to phase 2 IFs (*p* < 0.01).

Protein digestibility values are displayed in Table 5. The AAS in this case was corrected for protein digestibility (PDCAAS) based on the amino acid requirements for children aged 0 to 12 months old. Even after correction, values remained lower than 1.0 for all phase 1 and phase 2 IFs, indicating that essential amino acid concentrations in the investigated IFs were under the recommended limits. Furthermore, almost all PDCAAS were higher in phase 1 compared to phase 2 IFs (*p* < 0.01).

## 4. Discussion

IFs aimed for children under the age of one are formulated from cow milk or from the milk of other animal species or their mixture, and/or eventually, other ingredients suitable for infant feeding [23]. Herein, several IFs formulated from cow milk were evaluated. Human and cow milk differ not only in the amount of proteins, but also their quality. Human milk protein contents range from 0.8 to 1.3 g·100 mL^−1^, varying throughout the lactation stage, while whole cow milk contains about 3.33 g·100 mL^−1^, about three-fold higher than human milk [24,25,26]. The current Brazilian legislation, which agrees with European Commission Directive 2006/141/EC recommendations, establishes a minimum protein content for phase 1 cow milk-derivative formulas of 1.8 g·100 kcal^−1^ and a maximum of 3.0 g·100 kcal^−1^, while protein concentrations in phase 2 formulas should range between 1.8 g·100 kcal^−1^ to 3.5 g·100 kcal^−1^ [27,28]. On the other hand, the FAO/WHO Codex Alimentarius, comprising 189 member countries and one European Union member, which establishes international food and nutrition standards and guidelines, recommends that the protein content of cow-milk derivative formulas should range between 1.8 to 3.0 g·100 kcal^−1^, not exceeding the maximum limit [7,23].

In the present study, whole cow milk contained higher total protein contents than both phase 1 and phase 2 IFs, demonstrating that IFs manufacturers are making an effort to reduce protein contents from raw material seeking similar protein compositions to human milk. Additionally, protein contents in samples ME1B, NC1A, NC1C, NN1A, NN1C, DM2B and DA1B were all in accordance with their labels. Considering the protein content limits established by Brazilian legislation and Directive 2006/141/EC [27,28] for IFs labels (g·100 kcal^−1^), our findings indicate that ME1C, DM1A and DM1B phase 1 IFs contained protein contents above recommendations, but not below the minimum limit. In addition, all phase 2 IFs contained protein concentrations within the limits established by Brazilian current legislation. However, when considering the limits established by the Codex Alimentarius, more than one batch of each brand of some phase 1 and phase 2 IFs exceeded the maximum protein recommendation of 3.0 g·100 kcal^−1^, namely ME1C, DM1A, DM1B phase 1 IFs and ME2B, ME2C, NC2C, NN2B, DM2A, DM2C, DA2A and DA2C phase 2 IFs (Appendix A).

Disagreements concerning the minimum and maximum protein content values bring attention to the lack of consensus of worldwide guidelines. According to Food and Drug Administration (FDA) guidelines, the maximum protein content in IFs is established as 3.5 g·100 kcal^−1^, similar to the Brazilian limit, while the European Commission has updated infant food legislation and now established that minimum and maximum protein contents should be reduced to 1.6 to 2.5 g·100 kcal^−1^, below the former levels of 1.8 to 3.0 g·100 kcal^−1^ [29].

Nutritional information on food labels aids consumers and professionals to select a balanced diet, contributing to reducing the incidence of unhealthy conditions associated with inadequate eating habits [30]. Non-compliance to nutritional content label statements is foreseen in the Brazilian RDC nº 360 December 2003, which establishes a 20% tolerance considering label and experimental evaluation divergences [31]. At least one batch of the ME1, NC1, NN1, DM1, DA1, ME2, NC2, NN2 and DA2 IFs did not meet the current legislation, as protein discrepancies between protein concentrations and labeled values were higher than 20%. Non-compliance with regulatory agency statements concerning IFs components can lead to infant health risks, as excessive protein intake can overload the metabolic capacity of young children, impairing liver and kidney function and increasing dehydration risks [32,33].

Breast milk or IFs are the recommended exclusive protein sources for newborns and infants up to six months of age. Thus, the quality, amounts, and conformation of IFs proteins should be strictly controlled in order to sustain adequate and safe infant growth, ensuring short and long-term health and proper development. The amount and quality of protein intake during the first two years of life have important effects on muscle protein development and linear growth, supporting optimal brain development and, consequently, better cognitive evolution, and ensuring the development of healthy and mature immune and digestive systems [11,12]. Protein restrictions lead to low levels of insulin-like growth factor 1 (IGF-1), compromising not only growth but also adipogenesis control. On the other hand, excess protein, such as CM protein concentrations, is associated with greater weight gain, higher body mass index (BMI), and risk of overweight or later obesity [34].

A broad multi-center randomized and controlled clinical trial conducted in several countries investigated the effects of protein intake on infant growth and adiposity, where healthy infants fed cow milk-formulas containing high protein concentrations ranging from 2.9 to 4.4 g·100 kcal^−1^ or low protein concentrations varying from 1.77 to 2.2 g·100 kcal^−1^ before and after the 5th month of life, respectively, were compared to a group of exclusively breastfed babies. The ingestion of high protein formulas resulted in greater weight in the first two years of life, although no growth effect was observed [35]. Weight gain during childhood promoted by excess protein intake is the result of increased insulin-hyper aminoacidemia levels which, in turn, stimulates IGF-1 secretion, increasing later risks concerning obesity and associated diseases, reinforcing the requirement for strict IFs protein adjustments [35,36,37]. Furthermore, a significant effect of protein type and quality on gene expression is also observed, especially those encoding IGF-1 and insulin-like growth factor-binding proteins 1 (IGFBP-1), both involved in whole-body protein synthesis, thus affecting growth and body composition [12].

Considering protein IFs content and the Koletzko trials [35], the consumption of high protein formulas over the recommended amounts can result in significant weight gains in the first two years of life. Thus, one can assume that the regular intake of the ME1C (3.2 g·100 kcal^−1^), DM1A (3.1 g·100 kcal^−1^) and DM1B (3.3 g·100 kcal^−1^) IFs investigated herein, containing the maximum recommended protein content by children under the age of one may increase the risk for obesity. Protein IFs content should preferentially be kept closer to the minimum limit, in order to manage childhood risk for overweight and obesity.

Poor nutrition during pregnancy is related to low nephron endowment, which may comprise a potential driver for hypertension and renal disease later in life [38]. Prematurity and low birth weight have been associated with smaller kidneys and lower nephron endowment, predisposing individuals to kidney disease and hypertension in adulthood [39,40]. These findings support the hypothesis that kidneys may be programmed while nephrogenesis takes place. As kidneys quickly increase in size and functional capacity during the first months of life, an appropriate nutritional intervention could exert long-term cardiovascular and kidney health effects [33]. Nevertheless, excessive protein intake is harmful to infants due to increased protein metabolite filtration, mainly urea, which leads to glomeruli hyperplasia and immature renal tubules, increasing kidney growth due to enhanced renal workload, evidenced by the urea/creatinine serum ratio in infants fed high protein formulas [38]. Additionally, abnormal kidney sizes seem to be a compensatory mechanism to allow for the excretion of high protein loads. Furthermore, increased body weight following high protein intake stimulates the secretion of IGF-1, leading to a transient increase in body organ size, such as kidneys. Fortunately, kidneys can return to their normal volume by simply ceasing excess protein intake [33].

Ensuring nutrient consistency and standardization in distinct batches of the same brand of IFs should be a major goal concerning the nutritional quality of these formulas, as these formulas may comprise the exclusive food source offered to some infants for over six months, and marketing IFs with unrecommended nutrient composition/concentration can lead to inadequate child development. In the present study, consistency among batches from the same manufacturer was observed for DM1/DM2, DA1/DA2, ME2, NC2 and NN2, while brands ME1, NC1 and NN1, exhibited a non-uniform composition with statistical protein content differences in at least one of the three tested lots (Appendix A). This may be due to differential milk composition, as even milk samples from the same species do not contain exactly the same macro and micronutrient contents, and milk composition can be influenced by animal breed, lactation period or diet [41,42], as well as the manufacturing process, due to different purification methods (membrane filtration vs. ion exchange) and/or thermal processing [42,43].

In general, mammal milk contains insoluble proteins and soluble whey proteins, which include smaller floating proteins associated with milk fat globule membranes (MFGM). Caseins comprise the majority of insoluble milk proteins and are found in micellar form as α_s1_-CN, α_s2_-CN, β-CN and κ-CN, while whey proteins, i.e., β-Lg, α-La, Lf, lysozyme, bovine serum albumin, immunoglobulins (Igs) and other smaller soluble proteins, present in colloidal form and in their glycosylated forms, compose the MFGM [44,45,46,47]. While the whey: casein proteins ratio in breast milk is about 90:10, in colostrum, and 60:40 in mature milk, this ratio reaches 20:80 in CM, indicating that CM is richer in casein than human milk [48]. Caseins are hard to digest, as they coagulate in the stomach under acidic pH and attack intestinal cells, hampering nutrient absorption and resulting in intestinal bleeding, diarrhea, anemia, cramps, allergies and weight gain impairments [49]. In addition, caseins delay amino acid release due to slow digestion. On the other hand, whey proteins are quickly digested and highly bioavailable [48,50]. Herein, the IFs evaluation by SDS-PAGE indicated α-CN, β-CN, κ-CN, α-La, β-Lg, Lf as the main proteins, with other, less abundant ones, also detected. The amount of whey proteins α-La, β-Lg and Lf was higher than caseins in almost all IFs brands, except for DA1, DA2 and ME2 (Figure 3).

Whey or casein CM protein profiles differ from human milk. Human milk contains most protein subunits, such as α_s1_-CN, β-CN κ-CN and γ-CN, except for the α_s2_-CN variant, and main caseins correspond to β-CN, a highly phosphorylated protein, which aids in calcium absorption and contributes to zinc absorption. CM, however, contains main casein subunits α_s1_-CN, α_s2_-CN, β-CN, κ-CN and γ-CN, with higher concentrations of α-CN and κ-CN [51,52,53,54]. Among CM whey proteins, β-Lg corresponds to about 50%, while α-La comprises about 20%, and Lf contributes to ≈ 4%. In contrast, β-Lg is not present in human milk, and is, therefore, considered a potential allergen present in CM. α-La is the major protein in human milk, reaching approximately ≈ 40% among whey proteins, and rich in essential amino acids, mainly tryptophan and cysteine, involved in lactose synthesis and playing a role as a mineral absorption facilitator [55,56]. Lf contributes to ≈ 25% of whey proteins in human milk and exhibits strong antimicrobial activity against a broad spectrum of bacteria, viruses, yeast, fungi, and parasites [56,57,58]. IFs should, therefore, mimic the overall proportion of casein and whey proteins and contain whey protein profiles similar to mature human milk.

The HPLC analyses carried out herein, an analytical method more sensitive and accurate than the SDS-PAGE technique, demonstrated that casein IFs contribution to total protein contents was higher compared to whey proteins. κ-CN was detected by HPLC in higher concentrations compared to SDS-PAGE, where it was apparently the least abundant protein. κ-CN may undergo thiol disulfide linkages with proteins containing disulfide bonds, such as α-La, β-Lg, BSA, IGs and Lf, when subjected to high temperatures, resulting in ҡ-CN aggregates that move slowly or remain stacked in the stacking gel during the electrophoresis run, altering correct concentration estimations [59]. Consequently, the caseins and whey protein contents detected in the investigated IFs are not in agreement with public health authority recommendations and are dissimilar to human milk composition, where whey proteins should be present in higher amounts compared to caseins, as they are better digested due to coagulation in the acidic stomach environment, and consequently, delay the release of free amino acids [60]. The ideal formula to fulfill infant needs should contain α-CN, β-CN and β-Lg in lower amounts to those detected in the investigated IFs, as the most common type of food allergen for infants and young children, β-casein A1, has been associated with a range of allergic diseases, including type 1 diabetes [49,60,61,62]. When β-casein A1, commonly found in CM in Europe, the United States, Australia, and New Zealand [63], is digested, it releases the β-casomorphin-7 peptide, which presents opioid and inflammatory properties and is considered an allergen. However, although β-casein A1 comprises the main allergen agent, other factors should also be considered.

SDS-PAGE can simultaneously identify caseins and whey proteins, although information obtained following IFs sample SDS-PAGE seems to be less accurate compared to urea-PAGE or native-PAGE for caseins and whey proteins, respectively. Caseins display similar molecular weights, differing only slightly from each other, resulting in inadequate SDS-PAG discrimination [64]. Reverse phase-HPLC, on the other hand, can improve protein detection sensitivity and accuracy and may be useful in establishing a standard approach to evaluate milk proteins in complex samples such as infant formulas [18,19].

The protein concentrations described herein indicate significant differences between IFs brands and batches. Cow milk differences can be explained by protein milk content and the relative concentrations of individual proteins present during the lactation period, cattle feeding or formula manufacturing, which can spoil heat-sensitive nutrients [17,65]. Whey proteins, including β-CN, are more susceptible to heat treatment, as this affects their 3D-conformations, leading to loss of functional and nutritional characteristics, reducing the biovailability of calcium and zinc and, thus, affecting child and adulthood health [66,67]. Similarly, α-La may lose its ability to bind to calcium and zinc ions, and thus, decrease the bioavailability of these nutrients. On the other hand, Lf tends to lose its antimicrobial activity following heat exposure [68].

The supply of essential amino acids in the first month of a child’s life is provided by the set of proteins found in breast milk and/or IFs that must meet physiological newborn needs [69]. Whey protein fractions are richer in essential amino acids compared to caseins and, therefore, the general amino acid profile found in cow milk differs from breast milk [70], corroborating the results reported herein. Babies fed IFs exhibit greater differences in plasma amino acids compared to breastfed infants, with tryptophan levels, in particular, lower in the former [25].

Recommendation of essential and semi-essential amino acid contents per 100 kcal of IFs have been established considering breast milk proteins as reference [27,28]. The degree of compliance between the concentrations described in the present study observed herein and the content established by Brazilian RDCs (in mg·kcal^−1^) reveal that most of the investigated IFs contain amino acid concentrations under established guidelines, except for threonine in ME1, ME2, DM1, DM2, DA1 and DA2, which were above 77 mg·kcal^−1^ (reference value), and methionine in NC1, NC2, NN1, NN2, DM1, DM2 and DA1, detected at 24 mg·kcal^−1^, higher than the reference limit. It is important to note that a lower supply of essential amino acids and non-compliance to recommendations in the starting IFs (phase 1) can be more harmful when compared to the follow-up IFs (phase 2), as IF phase 1 is the only feeding recommended in the first six months of age when breast milk is not available. Unlike the starting IFs, follow-up IFs are offered to complement the introduction of new proteins from vegetable and animal origins. Therefore, even if the follow-up IFs are below the recommended intake, other protein sources can compensate and fulfill recommended essential amino acids concentrations.

Most essential amino acids were detected in all IFs, although at lower concentrations compared to established guidelines, probably due to reduced concentration of α-La and Lf whey proteins, the best source of essential amino acids, as indicated by the HPLC analysis (Table 1 and Table 2). Essential IFs amino acid contents were lower than the reference values and exhibited tryptophan as the first limiting amino acid, corroborating other studies [10,69]. Essential amino acids were significantly different among IFs brands and batches, reflecting protein fraction variability (Table 3), with the relative concentration of each amino acid also depending on CM composition [26,71].

Essential and semi-essential amino acids supplies, digestibility, absorption and transport through the gastrointestinal tract influence infant growth, and are important for cell and tissue maintenance [23]. The absorption of free amino acids, dipeptides or tripeptides in the small intestine depends on the bioaccessibility of IFs proteins, that release these nitrogenous compounds following gastrointestinal digestion [72]. Digestion and absorption processes are highly integrated, dynamic and complex, regulated by both neural and hormonal controls and responsive to various stimuli in order to efficiently release nutrients required for body growth, maintenance and reproduction. In vitro digestibility bioassays are reproducible and can provide reliable digestibility estimates for a wide variety of food matrices [73]. In vitro bioaccessibility assays simulate the composition and physicochemical features of the gastric and small intestinal fluids including mineral salts, different compounds and enzymes, mimicking the physiological conditions found in the human body [72]. Good bioaccessibility was verified following simulating independent gastric and intestinal IFs digestions, with IVPD over 90%, in both phases, with only a discrete difference between them. Regarding the intestinal phase, no differences were observed in the bioaccessibility of phase 2 IFs. The digestibility of breast milk, due to its composition and the physical structure of proteins and fat globules, differs from CM digestibility [74]. Casein digestibility is around 85% in CM and 94% in breast milk, while whey proteins reach 97–98% [75]. Furthermore, the IVPDs% reported herein are comparable to previously reported values, ranging from 85–95%, higher than CM and near breast milk values [76]. Taken together, the results regarding gastric the simulated gastric and intestinal digestions confirmed that the proteins present in phase 1 and 2 IFs are adequately digested, supporting suitable infant absorption.

Assessing the protein quality of foodstuffs is important when considering nutritional infant benefits. In this regard, the AAS and PDCAAS are commonly employed to assess the protein quality of food matrices [23]. The AAS compares the content of each essential amino acid present in a protein source with a protein reference, while the PDCAAS is defined as the ratio between the content of the first limiting amino acid in the protein and the content of that amino acid in a reference protein, multiplied by digestibility. Concerning infant requirement estimates, it is assumed that the human milk of a healthy and well-nourished mother during the first six months of life is an ideal intake source. Thus, the quality of the protein assessed by the chemical score is based on the limiting essential amino acid, where values greater than 1.0 and 100% for AAS and PCDAAS, respectively, indicate that good quality protein, containing essential amino acids capable of meeting human needs. Furthermore, according to the FAO/WHO [23] and IOM [43], IFs proteins should comprise at least 80% of breast milk protein. Herein, AAS and PDCAAS values concerning essential amino acid concentrations in IFs were lower than in breast milk, indicating that the most essential amino acids are limiting and do not reach the minimum established thresholds. When the quality of IFs proteins formulated from unmodified CM was evaluated through the chemical amino acid score (AAS), and corrected by the PDCAAS, threonine was the only amino acid that achieved a good chemical score of 80% in all phase 1 IFs and in the DM2 phase 2 formula.

According to IFs composition protocols, essential and semi-essential amino acids can be added to improve formula protein quality, but in amounts just enough to fill this purpose [27,28]. The findings obtained herein indicate that the protein quality from the investigated IFs was much lower than recommended. Therefore, alternatives to improve the protein quality should be carried out to supplement IFs with free essential amino acids or proteins of high biological value, such as α-La [77]. IFs enrichment with bovine α-La has been conducted in many clinical studies, improving the plasmatic levels of essential amino acids, resulting in infant development closer to that of breastfed ones [78,79]. Bovine α-La is noteworthy as a high biological value protein, as its essential amino acids, especially tryptophan, cysteine and lysine, reach similar concentrations found in human breast milk [26,49,68].

It is important to note that studies concerning the amino acid scores of CM-based IFs are scarce. In the present study, protein composition and essential amino acid ratios were variable, and total protein contents reported on the investigated IFs labels are not adequate and do not reflect IFs quality, making it difficult for pediatricians and nutritionists to accurately compare between available commercial products.

## 5. Conclusions

Despite attempts to mimic the composition or performance of breast milk by IFs manufacturers, proteins present in IFs cannot be proven as exhibiting the same bioavailability as those found in human milk. All formulas displayed adequate protein bioaccessibility, with proteins almost entirely digested in the intestine, generating free amino acid, dipeptides and tripeptides available for absorption. Only some IFs brands exhibited total protein content in accordance with product labels. Total protein content varied between batches from the same manufacturer, which may result in negative effects in children that consume the same brand regularly over six months. In addition, most IFs displayed protein contents above the maximum limits established by Brazilian regulation, as well as the European Commission Directive 2006/141/EC and Codex Alimentarius (FAO and WHO).

Casein contents were higher than whey proteins in all IFs, presenting three different subunits and with K-CN as the most abundant. Concerning whey protein contents, Lf amounts were lower than in human milk, jeopardizing newborn defenses, as this protein exhibits strong antimicrobial activity. High β-Lg concentrations comprised a non-desirable component, as this protein, which is not present in human breast milk, can induce allergy in susceptible individuals.

Therefore, despite the advances noted in technological IFs processing in recent years leading to safer and more adequate IFs for children under the age of one, the investigated brands still displayed marked nutritional quality differences when compared to human breast milk, requiring continuous improvements to follow current international guidelines due to the major health effects associated to poor nutrition or obesity.

## Figures and Tables

**Figure 1 nutrients-13-03933-f001:**
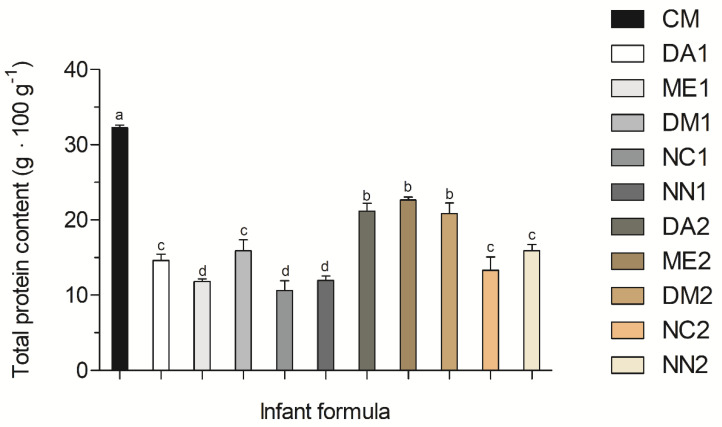
Protein content (g·100 g^−^^1^) of cow milk (CM) and phase 1 and phase 2 infant formulas. Formulas from the same brand were coded with the same two capital letters and numbers 1 or 2 to indicate the phase of infant growth. Three samples of each batch were analyzed and the values are expressed as means ± SD (*n* = 3). Different lowercase letters over the bars indicate differences between brands at a significance level of *p* < 0.05.

**Figure 2 nutrients-13-03933-f002:**
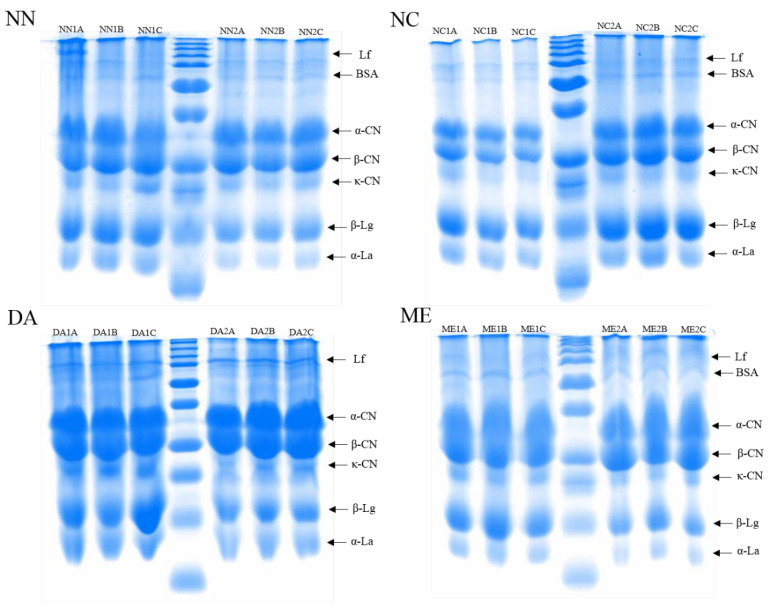
Protein profile of representative infant formulas resolved by SDS-PAGE 12%. Lanes 1, 2 and 3 of each gel represent phase 1 infant formulas (batches A, B and C), the central lanes correspond to Mw markers and lanes 5, 6 and 7 of each gel represent phase 2 infant formulas (batches A, B and C). Lf, lactoferrin; Mw, molecular weight; BSA, bovine serum albumin; α-CN, α-casein; β-CN, β-casein; κ-CN, κ-casein; β-Lg, β-lactoglobulin; α-La, α-lactalbumin. Electrophoresis was run at 200 V and 25 mA for 2 h. Gels were stained with a Coomassie Blue G-250 for 24 h and destained in acetic acid, methanol and distilled water (20:20:60) until the protein bands were visible for photo-documentation.

**Figure 3 nutrients-13-03933-f003:**
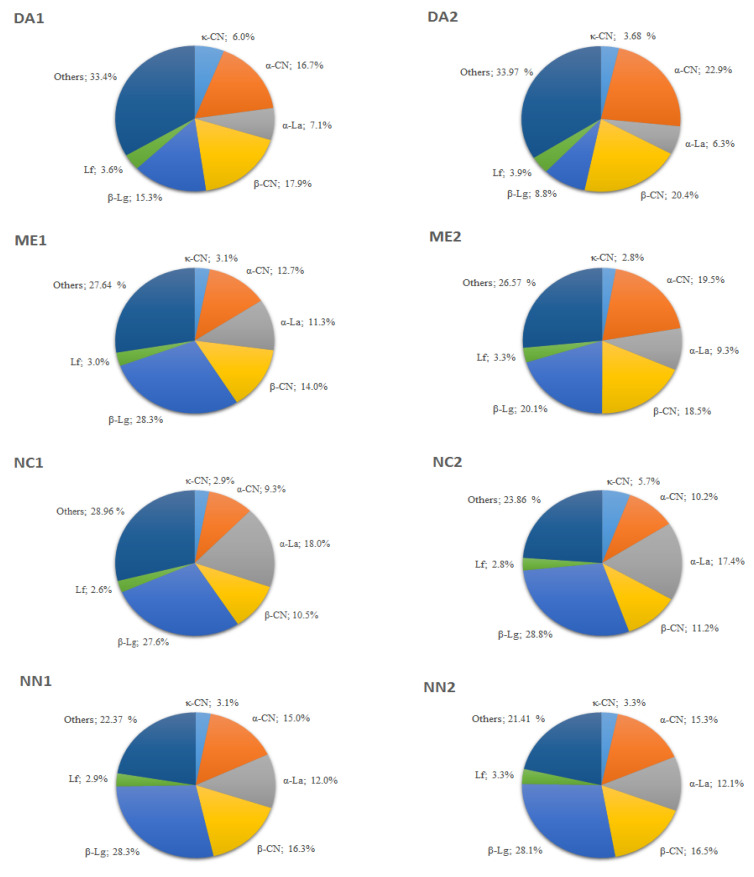
Approximate percentage of protein fractions identified by SDS-PAGE 12% for each infant formula. Lf, lactoferrin; BSA, bovine serum albumin; α-CN, α-casein; β-CN, β-casein; κ-CN, κ-casein; β-Lg, β-lactoglobulin; α-La, α-lactalbumin. Gels were analyzed by TotalLab Quant^®^ software, in which the apparent volume for each protein band was determined by densitometry. Ratios between whey proteins and caseins were expressed as a percentage using the apparent volume of protein fractions. Unidentified proteins were classified as other proteins.

**Figure 4 nutrients-13-03933-f004:**
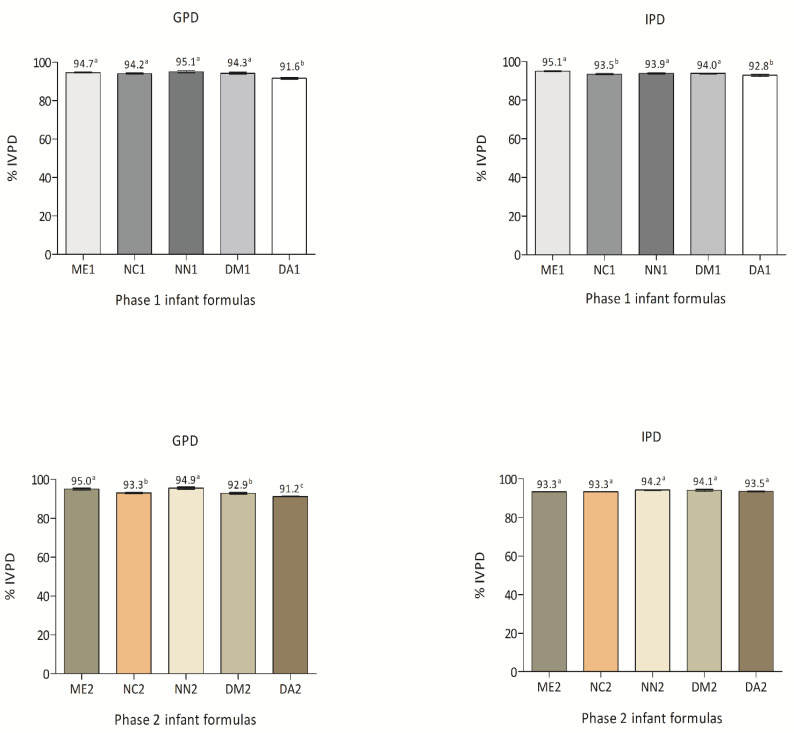
In vitro protein digestibility (%IVPD) of phase 1 and phase 2 infant formulas. Values are expressed as means ± SD (*n* = 3). Different letters denote difference at a significance level of *p* < 0.05. The percentage of IVPD was estimated based on the nitrogen content by the micro-Kjeldahl method.

**Table 1 nutrients-13-03933-t001:** Major protein contents in phase 1 and phase 2 infant formulas.

Infant Formulas	Major Proteins (mg·g^−1^)					
	Lf	α-CN	β-CN	κ-CN	β-Lg	α-La
CM	0.31 ± 0.02 ^e^	43.26 ± 0.30 ^a^	57.95 ± 0.50 ^a^	125.14 ± 0.39 ^a^	56.62 ± 0.41 ^a^	3.53 ± 0.13 ^c^
Phase 1						
ME1	1.37 ± 0.15 ^b^	14.77 ± 2.02 ^d,e^	7.96 ± 0.55 ^f^	100.50 ± 0.37 ^d^	22.20 ± 1.35 ^d,e^	0.33 ± 0.05 ^f^
NC1	0.57 ± 0.27 ^d^	21.17 ± 2.74 ^b^	11.13 ± 0.90 ^d^	89.90 ± 4.08 ^e^	19.37 ± 1.76 ^e^	12.10 ± 1.21 ^a^
NN1	0.85 ± 0.04 ^c^	7.60 ± 0.17 ^g^	11.61 ± 0.01 ^d^	124.50 ± 3.37 ^a^	39.83 ± 1.79 ^b^	13.23 ± 0.41 ^a^
DM1	1.37 ± 0.10 ^b^	11.17 ± 0.57 ^f^	25.16 ± 3.37 ^b,c^	100.23 ± 0.37 ^d^	15.77 ± 1.66 ^f^	3.93 ± 0.89 ^c,d^
DA1	1.13 ± 0.15 ^b^	12.57 ± 1.53 ^e,f^	7.76 ± 0.31 ^f^	106.53 ± 2.72 ^c^	14.10 ± 2.30 ^f^	3.40 ± 1.82 ^c,d^
Phase 2						
ME2	0.87 ± 0.07 ^c^	18.50 ± 0.81 ^c^	5.63 ± 1.06 ^g^	127.1 ± 3.90 ^a^	26.27 ± 3.17 ^c,d^	0.57 ± 0.11 ^f^
NC2	0.40 ± 0.11 ^e^	13.37 ± 1.12 ^d,e^	10.6 ± 0.11 ^e^	88.17 ± 7.85 ^e^	17.7 ± 1.75 ^e,f^	7.57 ± 0.45 ^b^
NN2	0.67 ± 0.15 ^c,d^	7.23 ± 0.21 ^g^	22.23 ± 3.62 ^c^	115.53 ± 2.21 ^b^	32.61 ± 3.91 ^c^	11.97 ± 0.87 ^a^
DM2	0.71 ± 0.20 ^c,d^	13.51 ± 0.88 ^d,e^	28.7 ± 3.96 ^b^	103.47 ± 3.03 ^c,d^	18.70 ± 2.66 ^e,f^	2.10 ± 1.12 ^d^
DA2	3.80 ± 3.01 ^a^	14.71 ± 2.55 ^d,e^	9.03 ± 1.10 ^f^	107.17 ± 1.14 ^c^	17.90 ± 2.08 ^e,f^	1.07 ± 0.11 ^e^

Cow milk and ten infant formulas marketed in Brazil were evaluated by HPLC analyses. Phase 1 formulas ME1, NC1, NN1, DM1 and DA1 and phase 2 formulas: ME2, NC2, NN2, DM2 and DA2 were analyzed in triplicate to evaluate major protein fractions expressed as means ± SD. Different letters superscript within the same column indicate significant differences between formulas at a significance level *p* < 0.01. CM, cow milk; Lf, lactoferrin; BSA, bovine serum albumin; α-CN, α-casein; β-CN, β-casein; κ-CN, κ-casein; β-Lg, β-lactoglobulin; α-La, α-lactalbumin.

**Table 2 nutrients-13-03933-t002:** Major proteins in phase 1 and phase 2 infant formulas evaluated in three distinct batches from the same manufacturer.

Infant Formulas	Major Proteins (mg·g^−1^)					
	Lf	α-CN	β-CN	κ-CN	β-Lg	α-La
Phase 1						
ME1A	1.51 ± 0.15 ^a^	16.62 ± 0.11 ^a^	7.62 ± 0.11 ^b^	100.21 ± 0.27 ^b^	20.61 ± 0.15 ^b^	0.31 ± 0.21 ^a^
ME1B	1.45 ± 0.11 ^a,b^	12.62 ± 0.22 ^c^	8.63 ± 0.21 ^a^	100.36 ± 0.27 ^b^	22.92 ± 0.15 ^a^	0.41 ± 0.11 ^a^
ME1C	1.21 ± 0.21 ^b^	15.11 ± 0.13 ^b^	7.75 ± 0.17 ^b^	100.95 ± 0.17 ^a^	23.02 ± 0.34 ^a^	0.39 ± 0.11 ^a^
NC1A	0.19 ± 0.17 ^a^	22.83 ± 0.15 ^a^	10.14 ± 0.27 ^b^	94.87 ± 0.29 ^a^	21.43 ± 0.11 ^a^	13.41 ± 0.23 ^a^
NC1B	0.22 ± 0.15 ^a^	18.02 ± 0.45 ^b^	11.57 ± 0.65 ^a^	86.07 ± 0.19 ^c^	18.45 ± 0.31 ^b^	11.42 ± 0.43 ^b^
NC1C	0.41 ± 0.26 ^a^	22.72 ± 0.54 ^a^	11.84 ± 0.46 ^a^	88.94 ± 0.48 ^b^	18.35 ± 0.41 ^b^	11.29 ± 0.22 ^b^
NN1A	0.73 ± 0.21 ^a^	7.82 ± 0.10 ^a^	11.65 ± 0.43 ^a^	121.25 ± 0.72 ^c^	41.94 ± 0.42 ^a^	13.18 ± 0.12 ^b^
NN1B	0.88 ± 0.32 ^a^	7.51 ± 0.12 ^b^	11.56 ± 0.22 ^a^	129.85 ± 0.42 ^a^	38.76 ± 0.21 ^b^	13.76 ± 0.32 ^a^
NN1C	0.99 ± 0.12 ^a^	7.54 ± 0.12 ^b^	11.76 ± 0.49 ^a^	122.55 ± 0.83 ^b^	38.97 ± 0.12 ^b^	12.95 ± 0.11 ^b^
DM1A	1.42 ± 0.15 ^a^	10.57 ± 0.53 ^b^	27.38 ± 0.34 ^b^	99.87 ± 0.11 ^b^	14.08 ± 0.42 ^c^	4.41 ± 0.19 ^a^
DM1B	1.43 ± 0.11 ^a^	11.56 ± 0.14 ^a^	29.19 ± 0.55 ^a^	100.57 ± 0.11 ^a^	16.09 ± 0.12 ^b^	4.51 ± 0.39 ^a^
DM1C	1.39 ± 0.13 ^a^	11.53 ± 0.23 ^a^	19.19 ± 0.91 ^c^	100.44 ± 0.31 ^a^	17.39 ± 0.83 ^a^	2.93 ± 0.26 ^b^
DA1A	1.33 ± 0.21 ^a^	13.55 ± 0.11 ^a^	7.54 ± 0.43 ^a^	108.33 ± 0.21 ^a^	13.25 ± 0.12 ^b^	2.21 ± 0.25 ^b^
DA1B	1.15 ± 0.12 ^a^	13.41 ± 0.21 ^a^	8.15 ± 0.25 ^a^	103.42 ± 0.21 ^b^	15.66 ± 0.13 ^a^	2.54 ± 0.15 ^b^
DA1C	1.01 ± 0.28 ^a^	10.81 ± 0.11 ^b^	7.74 ± 0.21 ^a^	107.98 ± 0.13 ^a^	13.52 ± 0.33 ^b^	5.54 ± 0.14 ^a^
Phase 2						
ME2A	1.06 ± 0.11 ^a^	19.22 ± 0.14 ^a^	6.81 ± 0.21 ^a^	129.09 ± 0.14 ^b^	22.63 ± 0.27 ^b^	0.58 ± 0.15 ^a^
ME2B	0.88 ± 0.22 ^a^	17.63 ± 0.12 ^c^	4.72 ± 0.11 ^c^	129.59 ± 0.14 ^a^	28.03 ± 0.18 ^a^	0.51 ± 0.15 ^a^
ME2C	0.87 ± 0.28 ^a^	18.77 ± 0.28 ^b^	5.42 ± 0.18 ^b^	122.56 ± 0.25 ^c^	28.24 ± 0.18 ^a^	0.76 ± 0.13 ^a^
NC2A	0.57 ± 0.19 ^a^	13.36 ± 0.26 ^b^	10.76 ± 0.14 ^a^	82.97 ± 0.41 ^c^	17.74 ± 0.29 ^b^	8.06 ± 0.24 ^a^
NC2B	0.39 ± 0.37 ^a^	12.45 ± 0.17 ^c^	10.54 ± 0.45 ^a^	97.25 ± 0.22 ^a^	19.35 ± 0.15 ^a^	7.66 ± 0.23 ^a^
NC2C	0.42 ± 0.14 ^a^	14.46 ± 0.63 ^a^	10.71 ± 0.15 ^a^	84.47 ± 0.23 ^b^	15.86 ± 0.16 ^c^	7.15 ± 0.24 ^b^
NN2A	0.61 ± 0.16 ^a^	7.09 ± 0.24 ^b^	25.61 ± 0.17 ^a^	113.17 ± 0.21 ^c^	28.54 ± 0.54 ^c^	12.57 ± 0.11 ^a^
NN2B	0.71 ± 0.15 ^a^	7.39 ± 0.25 ^a,b^	22.71 ± 0.28 ^b^	116.12 ± 0.43 ^b^	33.01 ± 0.11 ^b^	11.68 ± 0.21 ^b^
NN2C	0.73 ± 0.19 ^a^	7.47 ± 0.14 ^a^	18.41 ± 0.63 ^c^	117.43 ± 0.33 ^a^	36.32 ± 0.72 ^a^	11.83 ± 0.11 ^b^
DM2A	0.92 ± 0.17 ^a^	14.24 ± 0.26 ^a^	25.40 ± 0.23 ^c^	109.33 ± 0.15 ^a^	21.53 ± 0.51 ^a^	2.82 ± 0.22 ^a^
DM2B	0.94 ± 0.77 ^a^	13.81 ± 0.56 ^b^	27.62 ± 0.36 ^b^	100.96 ± 0.35 ^b^	18.46 ± 0.14 ^b^	2.71 ± 0.33 ^a^
DM2C	0.28 ± 0.21 ^b^	12.53 ± 0.14 ^c^	33.10 ± 0.39 ^a^	100.25 ± 0.14 ^c^	16.26 ± 0.24 ^c^	0.82 ± 0.52 ^b^
DA2A	8.55 ± 0.41 ^a^	12.51 ± 0.18 ^b^	7.93 ± 0.11 ^c^	108.54 ± 0.24 ^a^	19.37 ± 0.13 ^a^	1.02 ± 0.21 ^a^
DA2B	1.54 ± 0.21 ^b^	17.52 ± 0.25 ^a^	9.15 ± 0.12 ^b^	107.34 ± 0.11 ^b^	18.98 ± 0.15 ^b^	1.29 ± 0.11 ^a^
DA2C	1.44 ± 0.21 ^b^	14.11 ± 0.22 ^c^	10.13 ± 0.22 ^a^	105.74 ± 0.32 ^c^	15.59 ± 0.36 ^c^	1.09 ± 0.11 ^a^

Thirty different batches, three batches for each infant formula, marketed in Brazil were evaluated by HPLC analyses. Analyzes for each sample were performed in triplicate (*n* = 90) and data are reported as means ± SD. Different letters superscript within the same column indicate significant differences between infant formulas at a significance level *p* < 0.01. Lf, lactoferrin; BSA, bovine serum albumin; α-CN, α-casein; β-CN, β-casein; κ-CN, κ-casein; β-Lg, β-lactoglobulin; α-La, α-lactalbumin.

**Table 3 nutrients-13-03933-t003:** Average of essential amino acids contents from phase 1 and phase 2 infant formulas.

	Reference	Infant Formulas									
		Phase 1 (mg·g^−1^)					Phase 2 (mg·g^−1^)				
Amino Acids	FAO/WHO (mg·g of Protein^−1^) *	ME1	NC1	NN1	DM1	DA1	ME2	NC2	NN2	DM2	DA2
Threonine	31	3.93 ± 0.02 ^c,E^	3.08 ± 0.03 ^e,I^	3.18 ± 0.01 ^d,H^	4.19 ± 0.02 ^b,C^	4.34 ± 0.01 ^a,B^	4.07 ± 0.02 ^b,D^	3.24 ± 0.01 ^d,G^	3.04 ± 0.02 ^e,I^	4.70 ± 0.02 ^a,A^	3.72 ± 0.02 ^c,F^
Lysine	57	0.20 ± 0.04 ^a,A,B^	0.20 ± 0.03 ^a,A,B^	0.26 ± 0.03 ^a,A^	0.15 ± 0.01 ^b,C^	0.14 ± 0.01 ^b,C^	0.16 ± 0.01 ^b,C^	0.19 ± 0.01 ^a,B^	0.10 ± 0.01 ^c,D^	0.19 ± 0.01 ^a,B^	0.15 ± 0.01 ^b,C^
Histidine	20	0.15 ± 0.01 ^a,A^	0.14 ± 0.01 ^a,A^	0.13 ± 0.01 ^a,A^	0.14 ± 0.03 ^a,A^	0.14 ± 0.04 ^a,A^	0.12 ± 0.02 ^a,A^	0.14 ± 0.01 ^a,A^	0.16 ± 0.02 ^a,A^	0.16 ± 0.02 ^a,A^	0.11 ± 0.03 ^a,A^
Valine	43	0.33 ± 0.01 ^c,C^	1.23 ± 0.03 ^b,B^	1.27 ± 0.01 ^b,B^	1.34 ± 0.02 ^a,A^	0.27 ± 0.05 ^c,D^	0.35 ± 0.01 ^b,C^	1.29 ± 0.03 ^a,B^	1.27 ± 0.01 ^a,B^	1.25 ± 0.01 ^a,B^	0.26 ± 0.02 ^c,D^
Methionine	28	0.95 ± 0.01 ^e,F^	1.34 ± 0.01 ^c,D^	1.40 ± 0.02 ^b,C^	1.99 ± 0.01 ^a,B^	1.19 ± 0.01 ^d,E^	0.97 ± 0.01 ^d,F^	1.41 ± 0.01 ^b,C^	1.36 ± 0.01 ^c,D^	2.13 ± 0.02 ^a,A^	0.87 ± 0.02 ^e,G^
Isoleucine	32	0.94 ± 0.02 ^c,D^	0.90 ± 0.03 ^c,D^	0.90 ± 0.04 ^c,D^	1.69 ± 0.01 ^a,B^	1.32 ± 0.02 ^b,C^	0.96 ± 0.03 ^b,D^	0.90 ± 0.03 ^b,D^	0.81 ± 0.01 ^c,E^	1.99 ± 0.01 ^a,A^	0.91 ± 0.02 ^b,D^
Leucine	66	0.75 ± 0.04 ^d,G^	2.66 ± 0.04 ^c,E^	2.69 ± 0.01 ^c,E^	3.96 ± 0.02 ^b,C^	4.14 ± 0.02 ^a,B^	3.29 ± 0.02 ^b,D^	2.72 ± 0.02 ^c,E^	2.56 ± 0.01 ^d,F^	4.56 ± 0.01 ^a,A^	3.24 ± 0.03 ^b,D^
Phenylalanine	52	2.61 ± 0.01 ^b,C^	2.39 ± 0.01 ^c,E^	2.60 ± 0.03 ^b,C^	3.24 ± 0.04 ^a,A^	2.60 ± 0.03 ^b,C^	2.92 ± 0.01 ^b,B^	2.52 ± 0.01 ^d,D^	2.63 ± 0.02 ^c,C^	3.28 ± 0.02 ^a,A^	2.50 ± 0.02 ^d,D^
Tryptophan	8.5	0.14 ± 0.09 ^a,A^	0.13 ± 0.07 ^a,A^	0.12 ± 0.07 ^a,A^	0.13 ± 0.09 ^a,A^	0.11 ± 0.08 ^a,A^	0.11 ± 0.09 ^a,A^	0.10 ± 0.07 ^a,A^	0.14 ± 0.01 ^a,A^	0.12 ± 0.06 ^a,A^	0.14 ± 0.06 ^a,A^

Ten different infant formulas marketed in Brazil were analyzed by HPLC. Analyzes were performed in triplicate and data are reported as means ± SD. Different superscript lowercase letters within the same line indicate significant differences between infant formulas of the same phase at a significance level *p* < 0.001. Different superscript uppercase letters within the same line indicate significant differences between phase 1 and phase 2 infant formulas at a significance level *p* < 0.01. * Recommended amino acids daily intake for children aged 0 to 12 months (Joint WHO/FAO/UNU Expert Consultation, 2007).

**Table 4 nutrients-13-03933-t004:** Amino acid scores (AAS) for phase 1 and phase 2 infant formulas calculated for children aged 0–6 to 7–12 months.

Amino Acids	Phase 1 (mg·g^−1^)					Phase 2 (mg·g^−1^)				
	ME1	NC1	NN1	DM1	DA1	ME2	NC2	NN2	DM2	DA2
Threonine	1.011 ± 0.005 ^b,B^	1.019 ± 0.010 ^b,B^	0.970 ± 0.014 ^c,C^	1.008 ± 0.007 ^b,B^	1.169 ± 0.002 ^a,A^	0.655 ± 0.007 ^c,D^	0.595 ± 0.021 ^d,E^	0.644 ± 0.012 ^c,D^	0.890 ± 0.014 ^a,C^	0.665 ± 0.017 ^b,D^
Lysine	0.025 ± 0.003 ^a,A^	0.040 ± 0.003 ^a,A^	0.035 ± 0.007 ^a,A^	0.030 ± 0.014 ^a,b,A^	0.029 ± 0.002 ^b,B^	0.016 ± 0.003 ^b,C^	0.026 ± 0.001 ^a,B^	0.017 ± 0.002 ^b,C^	0.032 ± 0.008 ^a,B^	0.025 ± 0.005 ^a,B^
Histidine	0.058 ± 0.005 ^a,b,A^	0.065 ± 0.002 ^a,A^	0.045 ± 0.007 ^c,C^	0.050 ± 0.009 ^b,B^	0.059 ± 0.005 ^a,b,A^	0.030 ± 0.004 ^b,D^	0.035 ± 0.003 ^b,D^	0.020 ± 0.015 ^b,D^	0.050 ± 0.001 ^a,B^	0.032 ± 0.004 ^b,D^
Valine	0.065 ± 0.002 ^c,E^	0.261 ± 0.001 ^a,A^	0.215 ± 0.007 ^b,B^	0.205 ± 0.008 ^b,B^	0.053 ± 0.007 ^c,E^	0.040 ± 0.008 ^c,E^	0.145 ± 0.007 ^b,D^	0.185 ± 0.010 ^a,C^	0.170 ± 0.008 ^a,C^	0.039 ± 0.006 ^c,E^
Methionine	0.263 ± 0.001 ^c,D^	0.436 ± 0.013 ^a,A^	0.365 ± 0.007 ^b,B^	0.445 ± 0.008 ^a,A^	0.357 ± 0.003 ^b,B^	0.170 ± 0.001 ^d,F^	0.240 ± 0.007 ^c,E^	0.306 ± 0.019 ^b,C^	0.450 ± 0.001 ^a,A^	0.170 ± 0.001 ^d,F^
Isoleucine	0.228 ± 0.001 ^c,D^	0.255 ± 0.001 ^b,C^	0.210 ± 0.011 ^d,E^	0.340 ± 0.021 ^a,B^	0.345 ± 0.001 ^a,B^	0.140 ± 0.017 ^b,F^	0.135 ± 0.015 ^b,F^	0.159 ± 0.015 ^b,F^	0.370 ± 0.003 ^a,A^	0.160 ± 0.013 ^b,F^
Leucine	0.088 ± 0.001 ^d,G^	0.368 ± 0.001 ^b,C^	0.285 ± 0.008 ^c,D^	0.375 ± 0.020 ^b,C^	0.523 ± 0.003 ^a,A^	0.235 ± 0.010 ^d,E^	0.200 ± 0.007 ^e,F^	0.243 ± 0.001 ^c,E^	0.410 ± 0.003 ^a,B^	0.270 ± 0.008 ^b,D^
Phenylalanine	0.387 ± 0.007 ^b,B^	0.419 ± 0.023 ^a,A^	0.370 ± 0.010 ^b,B^	0.380 ± 0.007 ^b,B^	0.418 ± 0.001 ^a,A^	0.265 ± 0.006 ^c,D^	0.230 ± 0.001 ^d,E^	0.317 ± 0.029 ^b,C^	0.375 ± 0.011 ^a,B^	0.265 ± 0.008 ^c,D^
Tryptophan	0.133 ± 0.009 ^a,A^	0.145 ± 0.001 ^a,A^	0.105 ± 0.009 ^b,B^	0.115 ± 0.003 ^b,B^	0.112 ± 0.06 ^b,B^	0.070 ± 0.002 ^c,D^	0.067 ± 0.011 ^c,D^	0.111 ± 0.011 ^a,B^	0.082 ± 0.008 ^b,C^	0.091 ± 0.001 ^b,C^

Analyses were performed in triplicate and data are reported as means ± SD. Different superscript lowercase letters within the same line indicate significant differences between infant formulas of the same phase at a significance level *p* < 0.001. Different superscript uppercase letters within the same column indicate significant differences between phase 1 and phase 2 infant formulas at a significance level *p* < 0.01.

**Table 5 nutrients-13-03933-t005:** Protein digestibility-corrected by amino acid scores (PDCAAS) from phase 1 and phase 2 infant formulas calculated for children age 0–6 and 7–12 months.

Amino Acids	Phase 1 (mg·g^−1^)					Phase 2 (mg·g^−1^)				
	ME1	NC1	NN1	DM1	DA1	ME2	NC2	NN2	DM2	DA2
Threonine	1.010 ± 0.007 ^b,B^	0.965 ± 0.007 ^c,C^	0.967 ± 0.010 ^c,C^	0.995 ± 0.009 ^c,C^	1.115 ± 0.007 ^a,A^	0.595 ± 0.005 ^c,F^	0.575 ± 0.020 ^d,F^	0.580 ± 0.014 ^c,F^	0.850 ± 0.014 ^a,D^	0.635 ± 0.009 ^b,E^
Lysine	0.024 ± 0.007 ^b,B^	0.030 ± 0.005 ^a,A^	0.031 ± 0.002 ^a,A^	0.025 ± 0.004 ^b,A^	0.025 ± 0.003 ^b,B^	0.016 ± 0.004 ^b,B^	0.024 ± 0.003 ^b,B^	0.015 ± 0.005 ^b,B^	0.030 ± 0.003 ^a,A^	0.018 ± 0.004 ^b,B^
Histidine	0.055 ± 0.013 ^a,A^	0.063 ± 0.010 ^a,A^	0.044 ± 0.010 ^a,A^	0.045 ± 0.010 ^a,A^	0.055 ± 0.012 ^a,A^	0.027 ± 0.005 ^b,B^	0.030 ± 0.001 ^b,B^	0.015 ± 0.007 ^c,C^	0.045 ± 0.008 ^a,A^	0.030 ± 0.001 ^b,B^
Valine	0.061 ± 0.018 ^d,E^	0.255 ± 0.013 ^a,A^	0.210 ± 0.07 ^b,B^	0.180 ± 0.005 ^c,C^	0.051 ± 0.012 ^d,E^	0.038 ± 0.008 ^c,E^	0.135 ± 0.001 ^b,D^	0.175 ± 0.007 ^a,C^	0.165 ± 0.011 ^a,C^	0.035 ± 0.015 ^c,E^
Methionine	0.255 ± 0.011 ^c,D^	0.415 ± 0.010 ^a,B^	0.350 ± 0.07 ^b,C^	0.425 ± 0.007 ^a,A^	0.345 ± 0.001 ^b,C^	0.160 ± 0.002 ^d,E^	0.230 ± 0.015 ^c,D^	0.290 ± 0.010 ^b,D^	0.430 ± 0.004 ^a,A^	0.165 ± 0.005 ^d,E^
Isoleucine	0.225 ± 0.012 ^b,B^	0.245 ± 0.010 ^b,B^	0.180 ± 0.012 ^c,C^	0.319 ± 0.015 ^a,A^	0.325 ± 0.011 ^a,A^	0.137 ± 0.011 ^b,D^	0.133 ± 0.011 ^c,D^	0.150 ± 0.011 ^b,D^	0.350 ± 0.018 ^a,A^	0.150 ± 0.009 ^b,D^
Leucine	0.085 ± 0.001 ^d,G^	0.340 ± 0.008 ^b,C^	0.275 ± 0.011 ^c,D^	0.355 ± 0.011 ^b,C^	0.505 ± 0.005 ^a,A^	0.225 ± 0.008 ^b,E^	0.190 ± 0.14 ^c,F^	0.230 ± 0.001 ^b,E^	0.390 ± 0.005 ^a,B^	0.255 ± 0.015 ^b,D^
Phenylalanine	0.369 ± 0.014 ^c,B^	0.395 ± 0.013 ^b,B^	0.335 ± 0.011 ^d,C^	0.370 ± 0.009 ^b,B^	0.415 ± 0.003 ^a,A^	0.255 ± 0.013 ^c,D^	0.227 ± 0.017 ^d,D^	0.315 ± 0.018 ^b,C^	0.373 ± 0.011 ^a,B^	0.250 ± 0.007 ^c,D^
Tryptophan	0.125 ± 0.004 ^b,B^	0.142 ± 0.008 ^a,A^	0.101 ± 0.010 ^c,C^	0.080 ± 0.010 ^d,D^	0.105 ± 0.003 ^c,C^	0.069 ± 0.09 ^c,D^	0.065 ± 0.011 ^c,D^	0.105 ± 0.015 ^a,C^	0.080 ± 0.006 ^b,D^	0.085 ± 0.012 ^b,D^

Analyzes were performed in triplicate and data are reported as means ± SD. Different superscript lowercase letters within the same row indicate significant differences between infant formulas of the same phase at a significance level *p* < 0.001. Different superscript uppercase letters within the same column indicate significant differences between phase 1 and phase 2 infant formulas at a significance level *p* < 0.01.

## Data Availability

Data that supports the findings of these experiments are available upon request.

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
