# Peer review of "Protein Quality in Infant Formulas Marketed in Brazil: Assessments on Biodigestibility, Essential Amino Acid Content and Proteins of Biological Importance"

_nutrients, 2021, doi:10.3390/nu13113933_

Round 1

Reviewer 1 Report

This is an interesting study assessing the protein quality based on the essential amino acid content of starting and follow-up formulas from different manufacturers. The chemical amino acid score and protein digestibility corrected by the amino acid score were also assessed.

Some issues need to be addressed.

Major queries

  • Sample selection (line 103): it should be specified which criterion was used for infant formulas (IFs) selection: by convenience? Random? All available IFs in the market? The more used IFs?
  • In the Abstract it is stated that starting formulas and follow-up formulas were analyzed. However, in the body of the manuscript it seems that the authors have replaced these names with a terminology they chose: phase 1 formulas and phase 2 formulas. If this is the case, phase 1 formulas and phase 2 formulas must be defined in Methods.
  • Line170: the name of the nine essential amino acids analyzed should be specified.
  • Line 411: instead of “some older children” it should be “infants up to 6 months of age”
  • I suggest that In Discussion it should be emphasized that it is more problematic if the composition of essential amino acids of starting formulas does not comply with recommendations since they are conceived to be consumed as exclusive feeding within the first 6 months of age when breast milk is not available. In contrast, follow-up formulas are used after 6 months of age sharing the daily diet with complementary feedings which essential amino acids content may mitigate insufficiency or excess of formulas.

Minor queries

  • The abbreviation IFs was chosen for “infant formulas”. Therefore, wherever the abbreviation is used in the manuscript to mean infant formulas should be “IFs” and not “IF”
  • Line 64: I suggest replacing “consist” with “contain” since infant formulas consist of much more nutrients beyond proteins
  • Line 144: all abbreviations should be explained in full the first time they appear in the text
  • Line 288: the abbreviation RSD should be named in full
  • Line 327: what is meaning of “L”? Is it Lf?
  • Line 382: I suggest stating “aimed for children” instead of “aimed at children”
  • Lines 408-410 and 434-436: statements included in these lines need to be supported with at least with a reference
  • Line 443: “newborns” should be replaced with “infants”
  • Line 527: only the abbreviations AAS and PDCAAS should be used because they have been previously explained

Author Response

ANSWERS TO REVIEWER 1

Round 1 – Nutrients-1431959

Title: Protein quality in infant formulas marketed in Brazil: Assessments on biodigestibility, essential amino acid content and proteins of biological importance

GENERAL COMMENTS BY THE AUTHORS:

We believe that we have fully addressed all reviewer 1 concerns and comments. The criterion used for infant formulas (IFs) selection was improved for better understanding. We have emphasized the issue concerning the composition of essential amino acids of starting formulas that do not comply with recommendations in the discussion section. Furthermore, all minor requests specified by reviewer 1 were carried out to improve the quality of the manuscript.

The modifications suggested by the reviewer 1 have polished the manuscript and increased its overall impact. We would like to thank the reviewer for his/her insight and thoughtful critique of our manuscript. All

After the modifications suggested by the reviewers, the entire text was revised by an editing specialized company in order to improve English grammar and syntax. Reviewer comments precede our responses. Modifications were highlighted in yellow in the text.

General Comments by the REVIEWER 1:

This is an interesting study assessing the protein quality based on the essential amino acid content of starting and follow-up formulas from different manufacturers. The chemical amino acid score and protein digestibility corrected by the amino acid score were also assessed.

Major queries

  • Sample selection (line 103): it should be specified which criterion was used for infant formulas (IFs) selection: by convenience? Random? All available IFs in the market? The more used IFs?

Answer: As suggested by reviewer 1, the text concerning IF inclusion criteria was carefully revised (page 3, lines 105-118).

All IFs selected were marketed in several commercial establishments, conveniently available to be purchased by consumers in the most populated part of the metropolitan region of the municipality of Rio de Janeiro/Brazil. Although no formal inquiry was performed, as the selected formulas are commonly supplied in the largest supermarkets in town, they were considered a high demand by the consumers and evaluated in the present study.

All selected IFs brands were registered by the Brazilian regulatory agency (ANVISA), marketed as powdered formulas packed in aluminum cans and labeled by each manufacturer. Furthermore, all IFs met the following inclusion criteria: formulas should be prepared exclusively from non-hydrolyzed cow milk proteins, for children aged 0 to 6 months (phase 1 - starting IFs) and for children aged 6 to 12 months (phase 2 - follow-up IFs).

IFs produced using protein sources other than cow milk, such as soy or wheat protein, as well those designed for specific needs, such as lactose-free or hydrolyzed protein conditions, were not selected.

  • In the Abstract it is stated that starting formulas and follow-up formulas were analyzed. However, in the body of the manuscript it seems that the authors have replaced these names with a terminology they chose: phase 1 formulas and phase 2 formulas. If this is the case, phase 1 formulas and phase 2 formulas must be defined in Methods.

Answer:  This information was included in the abstract (page 1, lines 33-34) and introduction (page 2, lines 96-97), as suggested.

  • Line170: the name of the nine essential amino acids analyzed should be specified.

Answer: The reviewer is correct, and the suggestion was accepted by including  the name of the nine identified and quantified essential amino acids in the methods section (page 4, lines 177-178).

  • Line 411: instead of “some older children” it should be “infants up to 6 months of age”

Answer: This sentence has been modified, as suggested by the reviewer (page 15, line 453).

  • I suggest that In Discussion it should be emphasized that it is more problematic if the composition of essential amino acids of starting formulas does not comply with recommendations since they are conceived to be consumed as exclusive feeding within the first 6 months of age when breast milk is not available. In contrast, follow-up formulas are used after 6 months of age sharing the daily diet with complementary feedings which essential amino acids content may mitigate insufficiency or excess of formulas.

Answer: We agree with the reviewer’s suggestion and a sentence emphasizing the issue phase 1 formulas do not meet essential amino acid recommendations has been included, as they are designed to be consumed as exclusive feeding in the first 6 months of age when breast milk is not available (page 19, lines 548-552).

Minor queries

  • The abbreviation IFs was chosen for “infant formulas”. Therefore, wherever the abbreviation is used in the manuscript to mean infant formulas should be “IFs” and not “IF”

Answer: We agree with the reviewer suggestion and the abbreviation IFs were replaced by IFs in the entire manuscript, as suggested.

  • Line 64: I suggest replacing “consist” with “contain” since infant formulas consist of much more nutrients beyond proteins

Answer: We agree with the reviewer’s suggestion and the word consist was replaced by contain in page 2, line 64, as suggested.

  • Line 144: all abbreviations should be explained in full the first time they appear in the text

Answer: The reviewer is correct. The entire text was revised in order to explain all abbreviations in full the first time they appear in the text, as suggested.

  • Line 288: the abbreviation RSD should be named in full

Answer: We agree with the reviewer’s corrections and the name in full was added before the abbreviation RSD (page 8, line 298), as suggested.

  • Line 327: what is meaning of “L”? Is it Lf?

Answer: The reviewer is correct. “L” was replaced by Lf (page 9, line 335).

  • Line 382: I suggest stating “aimed for children” instead of “aimed at children”

Answer: The expression “aimed at children” was replaced by “aimed for children” as suggested by the reviewer (page 14, line 424).

  • Lines 408-410 and 434-436: statements included in these lines need to be supported with at least with a reference

Answer: The reviewer is correct. Two additional references were included in these lines, as suggested (page 15, lines 452 and 466).

  • Line 443: “newborns” should be replaced with “infants”

Answer: The word “newborns“ was replaced by “infants”, as suggested by the reviewer (page 16, line 485).

  • Line 527: only the abbreviations AAS and PDCAAS should be used because they have been previously explained

Answer: As suggested by the reviewer, only the abbreviations AAS and PDCAAS were used page 19, line 572.

Reviewer 2 Report

In the reviewed publication on the quality of protein in formulas commercially available in Brazil, intended for 1-year-old children, the authors present a comprehensive assessment of the quantitative-qualitative composition of the protein in these products. The total protein content, the composition of its fractions, as well as the content of exogenous amino acids in relation to the golden standard which is the composition of breast milk, protein bioavailability and compliance with law regulations were determined.

The authors of the study tested  products / samples (n=10/30) from various manufacturers (n=3) of infant formula and follow-on formula. They measured the total protein content, identified the basic protein fractions - caseins and whey proteins. They compared them with the composition of human milk and cow's milk. The study showed that most of the tested infant products had a higher protein content than the FAO/WHO guidelines. There is 3 times less protein in human milk than in cow's milk, which is the basis for the production of formulas and follow-on formulas, which is the basis for the nutrition of infants who are not breastfed. Whey and casein proteins in human milk are present in the ratio of 60:40. Whey proteins, such as lactoferrin, lysozyme, secreted IgA, have limited nutritional properties due to resistance to proteolytic enzymes, but play an important role in immune mechanisms. The main nutritional whey protein is alpha-lactalbumin. All tested products contained lactoferrin and alpha-lactalbumin below recommended concentrations, whereas kappa-casein, alpha-casein and beta-lactoglobulin where above recommended concentrations.

The authors also identified the necessary amino acids and determined their content in the tested samples. They found that the most abundant amino acids were threonine, leucine, and tryptophan. In mature breast milk it is leucine (94.5 mg/100ml), lysine (66.8 mg/100ml), valine (54.8 mg/100ml), isoleucine (51.6 mg/100ml) [Zhang, 2013, Nutrients, 5: 4800-4821]. The amino acid composition of the formula and follow-on formula should be similar to the amino acid composition of human milk proteins.

The publication presented for review draws attention to the need of:

  1. constant control/monitoring of the composition of ready to eat food/formula intended for infants and young children,
  2. continuation of research towards the optimization of protein supply in the diet of non-breastfed infants.

In my opinion, this publication is of great importance not only for doctors and nutritionists in terms of promoting breastfeeding, but also for manufacturers of infant formulas and follow-on formulas in order to standardize the quantitative and qualitative composition of this food.

I have no comments on the methodology of work, analytical and statistical methods, presentation of results - tabular and graphic form.

Author Response

ANSWERS TO REVIEWER 2

Round 1 – Nutrients-1431959

Title: Protein quality in infant formulas marketed in Brazil: Assessments on biodigestibility, essential amino acid content and proteins of biological importance

GENERAL COMMENTS BY THE AUTHORS:

We would like to thank the reviewer for his/her insight and thoughtful critique of our manuscript. We are very glad we have reached a clear and meaningful written manuscript. The importance of the content of essential amino acids in phase 1 IF to reach recommended concentrations were better explained, as phase 1 formulas are the only feeding offered to infants at 0-6 months. The inclusion criteria for infant formulas were rewritten for clarification. Several expressions were replaced to improve meaning. Modifications were highlighted in yellow in the revised text.

After all modifications, the entire text was revised by an editing specialized company in order to improve English grammar and syntax.

General Comments by the REVIEWER 2:

In the reviewed publication on the quality of protein in formulas commercially available in Brazil, intended for 1-year-old children, the authors present a comprehensive assessment of the quantitative-qualitative composition of the protein in these products. The total protein content, the composition of its fractions, as well as the content of exogenous amino acids in relation to the golden standard which is the composition of breast milk, protein bioavailability and compliance with law regulations were determined.

The authors of the study tested products / samples (n=10/30) from various manufacturers (n=3) of infant formula and follow-on formula. They measured the total protein content, identified the basic protein fractions - caseins and whey proteins. They compared them with the composition of human milk and cow's milk. The study showed that most of the tested infant products had a higher protein content than the FAO/WHO guidelines. There is 3 times less protein in human milk than in cow's milk, which is the basis for the production of formulas and follow-on formulas, which is the basis for the nutrition of infants who are not breastfed. Whey and casein proteins in human milk are present in the ratio of 60:40. Whey proteins, such as lactoferrin, lysozyme, secreted IgA, have limited nutritional properties due to resistance to proteolytic enzymes, but play an important role in immune mechanisms. The main nutritional whey protein is alpha-lactalbumin. All tested products contained lactoferrin and alpha-lactalbumin below recommended concentrations, whereas kappa-casein, alpha-casein and beta-lactoglobulin where above recommended concentrations.

The authors also identified the necessary amino acids and determined their content in the tested samples. They found that the most abundant amino acids were threonine, leucine, and tryptophan. In mature breast milk it is leucine (94.5 mg/100ml), lysine (66.8 mg/100ml), valine (54.8 mg/100ml), isoleucine (51.6 mg/100ml) [Zhang, 2013, Nutrients, 5: 4800-4821]. The amino acid composition of the formula and follow-on formula should be similar to the amino acid composition of human milk proteins.

The publication presented for review draws attention to the need of: constant control/monitoring of the composition of ready to eat food/formula intended for infants and young children, continuation of research towards the optimization of protein supply in the diet of non-breastfed infants.

In my opinion, this publication is of great importance not only for doctors and nutritionists in terms of promoting breastfeeding, but also for manufacturers of infant formulas and follow-on formulas in order to standardize the quantitative and qualitative composition of this food.

I have no comments on the methodology of work, analytical and statistical methods, presentation of results - tabular and graphic form.
